# Biodegradable and Biocompatible Thermoplastic Poly(Ester-Urethane)s Based on Poly(ε-Caprolactone) and Novel 1,3-Propanediol Bis(4-Isocyanatobenzoate) Diisocyanate: Synthesis and Characterization

**DOI:** 10.3390/polym14071288

**Published:** 2022-03-23

**Authors:** Alejandra Rubio Hernández-Sampelayo, Rodrigo Navarro, Dulce María González-García, Luis García-Fernández, Rosa Ana Ramírez-Jiménez, María Rosa Aguilar, Ángel Marcos-Fernández

**Affiliations:** 1Institute of Polymer Science and Technology (CSIC), Juan de la Cierva, 3, 28006 Madrid, Spain; alejandra@ictp.csic.es (A.R.H.-S.); luis.garcia@csic.es (L.G.-F.); raramirez@ictp.csic.es (R.A.R.-J.); mraguilar@ictp.csic.es (M.R.A.); 2Universidad Nacional de Educación a Distancia (UNED), Facultad de Ciencias, C/Bravo Murillo, 38, 28015 Madrid, Spain; 3Instituto Politécnico Nacional, Escuela Superior de Ingeniería Química e Industrias Extractivas, UPALM-Zacatenco, Col Lindavista, Mexico City 07738, Mexico; dgonzalezg@ipn.mx; 4Universidad de Guanajuato, Departamento de Química, Noria Alta s/n, Guanajuato 36050, Mexico; 5Biomedical Research Networking Center in the Subject Area of Bioengineering, Biomaterials and Nanomedicine (CIBER-BBN), Avenida Monforte de Lemons 3–5, 28029 Madrid, Spain

**Keywords:** biodegradability, biocompatibility, thermoplastic polyurethane, non-toxic polyurethane, hydrolyzable chain extender

## Abstract

A series of non-toxic biodegradable and biocompatible polyurethanes bearing p-aminobenzoate moieties are presented. The introduction of this attractive motif was carried out by the synthesis of a novel isocyanate. These biodegradable polymers were chemically and physically characterized by several techniques and methods including bioassay and water uptake measurements. The molecular weight of the soft segment (poly-ε-caprolactone, PCL) and hard segment crystallinity dictated the mechanical behavior and water uptake. The behavior of short PCL-based polyurethanes was elastomeric, whilst increasing the molecular weight of the soft segment led to plastic polyurethanes. Water uptake was hindered for long PCL due to the crystallization of the soft segment within the polyurethane matrix. Furthermore, two different types of chain extender, hydrolyzable and non-hydrolyzable, were also evaluated: polyurethanes based on hydrolyzable chain extenders reached higher molecular weights, thus leading to a better performance than their unhydrolyzable counterparts. The good cell adhesion and cytotoxicity results demonstrated the cell viability of human osteoblasts on the surfaces of these non-toxic biodegradable polyurethanes.

## 1. Introduction

The versatility of polyurethanes (PUs) becomes patent in the wide range of industrial sectors where they are commonly used, due to the possibility of tailoring their physico-chemical properties by careful selection of their components. Typically, PUs are built by the reaction of a macroglycol, a polyisocyanate and a chain extender. The macroglycol chains form the so-called soft segment (SS), while the reaction between the isocyanate and the chain extender produces the hard segment (HS). Depending on the balance and thermodynamic incompatibility between these types of segments, the polyurethane matrix can exhibit a characteristic and attractive phase separation.

In the biomedical area, the use of polyurethanes is widespread, even exceeding other polymeric materials (such as natural rubber or PVC), due to their ability to mimic the behavior of different tissues and show relatively good biocompatibility. The vast majority of polyurethanes are designed to withstand long periods of service, which require biostable materials [1]. However, there is currently great interest in the development of biodegradable polymers for certain biomedical applications, such as tissue regeneration [2,3] or controlled drug release [4,5]. Specifically, in these applications, PUs not only need to meet certain physical and/or mechanical requirements, but it is also necessary to assess whether the building blocks and their degradation products are biocompatible, non-toxic, and metabolized by the living organism. Therefore, polyurethanes obtained from the classical aromatic isocyanate 4,4′-diisocyanatodiphenyl-methane (MDI) or poly(urethane-urea)s with aromatic diamines (such as 3,3′-dichloro-4,4′-diaminodiphenylmethane, MOCA) as chain extenders, have been discarded because of the carcinogenic character of the diamines obtained as degradation products [6].

To overcome these drawbacks, several alternatives have been proposed. One alternative is the use of biodegradable chain extenders, such as amino acid derivatives, [7,8,9] ester-diols [10,11] or amide-diols such as N,N′-ethylene-bis(6-hydroxycaproamide) (EDA-2CL) [12]. In these cases, the introduction of labile functional groups such as esters, within the chemical structure of these alternative extenders, facilitates the degradation of the hard segment formed, without compromising the biological toxicity of the polyurethane. On the other hand, the alternative to aromatic diisocyanates is to use aliphatic diisocyanates such as 1,6-hexamethylenediisocyanate (HDI) [13] or diisocyanates of amino acids (mainly lysine) [14,15,16], which degrade towards non-toxic or slightly toxic diamines. However, these alternatives present new challenges that hinder the development of high-performance polyurethanes. In this regard, for example, aliphatic diisocyanates are characterized by a reduction in reactivity compared to their aromatic counterparts [17]. Another difficulty is related to the asymmetry of amino acids, which prevents the correct crystallization of the hard segment and subsequent phase separation within the polyurethane matrix. Consequently, the mechanical properties of the synthesized polyurethanes were mainly governed by the crystallization of the soft segment.

As a step forward, we intend to synthesize non-toxic biodegradable polyurethanes from an aromatic diisocyanate without adverse effects. This novel diisocyanate is based on the diamine 1,3-propanediol bis(4-aminobenzoate) (diamino-PABA), leading to a highly reactive and symmetrical aromatic diisocyanate. The precursor diamine has been rather scarcely used as a chain extender, and no diisocyanate derived from it has been described to date [17,18]. Therefore, in the present work, we describe the synthesis of a novel aromatic isocyanate derived from diamino-PABA and the synthesis and characterization of linear polyurethanes obtained from it. Polycaprolactones (PCLs) of different molecular weights were used as soft segments and, 1,4-butanediol (BD), which is not hydrolyzable, and EDA2-CL which is hydrolyzable, as chain extenders. Finally, due to the interesting features of these materials, cell adhesion and proliferation studies have also been investigated using fibroblasts and osteoblast cells.

## 2. Materials and Methods

### 2.1. Materials

1,3-Propanediol bis(4-aminobenzoate) (diamino-PABA), 1,2-ethylenediamine (EDA), stannous 2-ethylhexanoate (Sn(Oct)_2_), and triphosgene were purchased from Sigma-Aldrich, Madrid, Spain, and were used without further purification. Polycaprolactone diols with nominal molecular weights of 530, 1250 and 2000 g·mol^−1^, were also purchased from Sigma-Aldrich, Madrid, Spain. The molecular weight of the PCLs was accurately determined by proton NMR [8] and the values obtained were 519 (PCL519), 1100 (PCL1100) and 2054 (PCL2054) g·mol^−1^. The PCL diols were vacuum-dried at 70 °C for at least 5 h and were stored in a desiccator under vacuum until used.

4,4′-Diphenylmethane diisocyanate (MDI) and ε-caprolactone (ε-CL) were kindly donated by Lubrizol Advanced Materials Spain SL, Montmeló, Spain. The MDI was purified by short path distillation at 110 °C and 0.4 mbar, using a vacuum sublimator connected to a cryostat at −15 °C. The sublimated MDI was stored in a desiccator under vacuum until used. This purified MDI was used within 10 days due to its tendency to dimerize.

N, N-Dimethylacetamide (DMAc), purchased from Scharlau (Barcelona, Spain), was vacuum-distilled over commercial polymeric MDI-isocyanate, to remove residual water and amines that would lead to undesired drift in the stoichiometry on the synthesis of polymers.

### 2.2. Characterization

Proton (^1^H) and Carbon (^13^C) nuclear magnetic resonance (NMR) spectra were recorded on the Bruker spectrometer Oxford 400, at room temperature in deuterated solvents (CDCl_3_ and DMSO-d_6_). NMR spectra were referenced to the residual peak of these deuterated solvents (7.26 ppm and 77.0 ppm for CDCl_3_ and 2.50 ppm and 39.5 ppm for DMSO-d_6_).

The number and weight-average molecular weights and the polydispersity index were determined by size exclusion chromatography (SEC) using a Waters apparatus (Waters Division Millipore, Madrid, Spain), equipped with a refractive index detector. A set of Styragel HR3 and HR5 Waters columns (300 × 7.8 mm, 5 μm nominal particle size) conditioned at 70 °C was used to elute the samples at a 0.7 mL min-1 flow rate. The mobile phase was N, N-dimethylformamide (DMF) with 0.1% of LiBr. Polystyrene standards (Polymer Laboratories) used for calibration.

Fourier transform infrared spectra were recorded using a Perkin-Elmer spectrometer, model Spectrum One (Perkin-Elmer, Waltham, MA, USA), coupled with an attenuated total reflection (ATR) accessory. Sixteen scans were averaged from 4000 to 450 cm^−1^ and with a resolution of 2 cm^−1^.

Thermogravimetric analysis (TGA) was carried out in a Mettler Toledo TGA/SDTA 851 instrument (Mettler-Toledo, Schwerzenbach, Switzerland). Disc samples cut from films were heated from room temperature to 600 °C under a nitrogen atmosphere at a 10 °C min^−1^ heating rate.

Differential scanning calorimetry (DSC) was performed in a Mettler Toledo 822e calorimeter equipped with a liquid nitrogen accessory. Disc samples weighing approximately 10–15 mg were sealed in aluminum pans. The samples were initially heated from 25 to 90 °C with a heating rate of 10 °C min^−1^, then cooled to −90 °C at the maximum rate of the instrument, maintained for 5 min at this temperature and re-heated from −90 to 200 °C at 10 °C min^−1^. The glass transition temperature (Tg) was taken as the mid-point of the transition, whereas melting points (Mp) were taken as the maximum of the endothermic transition.

Tensile properties were measured in an MTS Synergie 200 testing machine equipped with a 100 N load cell. Test specimens were cut with the dimensions established in standard ISO37 (Type 4). A cross-head speed of 5 mm min^−1^ was used and the strain was measured from cross-head separation and referred to a 10 mm initial length. For all the synthetized polymers, a minimum of 5 specimens were analyzed.

### 2.3. Synthesis of 1,3-Propanediol Bis(4-Isocyanatobenzoate) (IsoPABA)

In a 250 mL three-necked round bottom flask a solution of 15.34 mmol of triphosgene with 15 mL of anhydrous toluene was cooled at 0 °C with an ice bath. A solution of 16.30 mmol of diamino-PABA in 60 mL of toluene was added dropwise to the previous solution. During the addition, the reaction temperature was kept below 5 °C. After the addition was complete, the reaction mixture was kept for one hour at 0 °C and subsequently heated at 110 °C for 10 h. Finally, the solvent was removed under reduced pressure with a rotary evaporator. The dry reaction crude was subsequently purified by sublimation, heating at 190 °C and reduced pressure (0.42 mbar). Yield 56%.

### 2.4. Synthesis of N, N’-Ethylene-Bis(6-Hydroxycaproamide) (EDA-2CL)

The synthesis of this chain extender has been described previously [12]. In a typical run, 0.2 mol of 1,2-ethylenediamine (EDA) and 2.0 mol of ε-caprolactone (ε-CL) were heated in an oil bath at 70 °C for 2 h. After cooling, the desired product precipitated as a white solid, which was isolated by filtration and exhaustively washed with toluene. The solid was then dried in a vacuum (yield 25%). The synthetic route is described in Figure 1.

### 2.5. Synthesis of Hard Segment Models

As an example, the reaction between isoPABA and EDA-2CL is described. In a 25 mL round-bottom flask, an equimolar mixture of isoPABA and EDA-2CL was dissolved in anhydrous DMAc (4 mL) and 2 drops (12 mg) of catalyst (Sn(Oct)_2_) were added. This mixture was immersed in a pre-heated oil bath at 80 °C. The reaction crude was stirred for 3 h at this temperature and at room temperature overnight. The crude reaction was poured over cold water and the solid was filtered and dried under vacuum. The same procedure was followed for the hard segment model isoPABA-BD.

### 2.6. Synthesis of Segmented Poly(Ester-Urethane)s

The composition of these segmented polymers was set at 50% by weight of hard segment, defined as ((weight of isoPABA + weight of extender)/total weight) × 100. The method for obtaining these polymers consists of two stages: first, the formation of a prepolymer, followed by the addition of the chain extender. In a reaction flask, isoPABA and dry PCL diol in the appropriate ratio and anhydrous DMAc were charged to produce a 50:50 (*w*/*v* = weight of reactants in gram/volume of DMAc in mL) mixture. Then, two drops of the catalyst (stannous 2-ethylhexanoate) were added. The reaction mixture was blanketed with N_2_ and immersed in a silicone bath preheated at 80 °C. After 3 h stirring at that temperature, the corresponding amount of chain extender (BD or EDA-2CL) was added. Heating was continued for a further three hours at 80 °C and then the reaction was warmed to room temperature and stirred overnight. The resulting viscous solution was cast onto a levelled heating plate and the solvent was slowly evaporated by heating at 60 °C. Once the solvent was removed, a homogeneous polymer film was obtained, which was stored under vacuum to remove the residual solvent.

### 2.7. Hydrolytic and Enzymatic In Vitro Degradation

In vitro hydrolytic degradation of the PUs was evaluated following the weight changes at certain time intervals in films submerged in phosphate buffer solution at 37 °C. The water uptake was taken from the maximum of the curve and the weight change was calculated from the equation:(1)$$W_{Change}\left(\%\right)=100*\left(W_{m}-W_{0}\right)/W_{0}$$
where *W_m_* is the weight of the hydrated specimen and *W*_0_ is the initial weight of the specimen. At least three specimens were tested.

In vitro enzymatic degradation was evaluated using porcine liver esterase (PLE) (Sigma-Aldrich, Madrid, Spain). The enzymatic solution was prepared by dissolving the PLE at 30 U/mL in 0.05 M phosphate buffered saline (PBS Dulbecco’s); the solutions were filtered using a 0.22 µm filter. Polyurethane discs were exposed to PLE in triplicate by immersion in the PLE solution. Enzyme activity was maintained by replacing the PLE solution every 24 h. The changes in weight of the polyurethanes were calculated using Equation (1).

### 2.8. Cytotoxicity Test

The materials and the discs of the negative control, Thermanox^®^ (TMX, Labclinics SA), were immersed in 5 mL of fresh and sterile medium, in the presence and in the absence of PLE, and kept under stirring at 37 °C. After 2, 7, 14, 21 and 28 days, the medium was removed (leachates) and replaced with fresh medium.

Due to the possible use of these materials for osteochondral regeneration, human osteoblast cells (HObs, Innoprot) and articular human chondrocytes (HC-a, Innoprot) were cultured using fresh complete culture medium. HObs and HC-a were seeded in 96-well plates at a concentration of 1.1 × 10^5^ cells/mL and maintained for 24 h at 37 °C with 5% CO_2_. After 24 h, the culture medium was exchanged with the corresponding leachates (n = 8), and the plates were incubated for 24 h under the same conditions. After this time, the contents of the wells were replaced by the MTT (3-[4,5-dimethylthiazol-2-yl]-2,5-diphenyltetrazoliumbromide) solution (10% in the fresh medium). The MTT reagent was kept in contact with the cultures for 4 h at 37 °C, and then, the contents of the wells were extracted and 100 μL of dimethylsulfoxide (DMSO) was added, in order to dissolve the formazan crystals that may have formed. The optical density was read at 570 nm with a reference wavelength of 630 nm in a Biotek Synergy HT plate reader. The relative cell viability (% *CV*) was calculated with respect to the control, from Equation (2):(2)$$\%\;CV=\frac{OD_{S}-OD_{B}}{OD_{C}}\cdot 100$$
where *OD_S_*, *OD_B_* and *OD_C_* are the optical density measurements of the sample, the blank and the control, respectively.

### 2.9. Cell Adhesion and Proliferation Assay

Cell adhesion and proliferation were measured by an Alamar Blue assay at 1, 7, 14 and 19 days, using HOb and HC-a cells.

The different PUs were directly deposited on a 24-well plate. HOb and HC-a cells were seeded on the PUs at a concentration of 1.4 × 10^5^ cells/mL in fresh medium. After the stablished time intervals, the medium was removed and a 10% solution of Alamar Blue in fresh medium, without phenol red, was added and the samples were incubated for 4 h. After this time, the medium was transferred to a 96-well plate and the fluorescence was measured at 570/630 (em/ex) in a microplate reader (Biotek Synergy HT spectrophotometer).

## 3. Results and Discussion

### 3.1. Synthesis of Aromatic Isocyanate IsoPABA

The biological interest of this aromatic isocyanate (isoPABA) lies in the structural analogy that it presents with respect to p-aminobenzoic acid. In fact, its chemical structure is very close to the local pain reliever benzocaine. In biological tests it has been shown that p-aminobenzoic esters show a very short half-life. These aromatic esters are rapidly metabolized and the corresponding metabolites are expelled through the urine [19]. Additionally, its symmetric chemical structure could favor the phase crystallization and/or phase separation of the hard segment within the polyurethane matrix, leading to an improvement in its physical performance. Furthermore, the introduction of ester groups into the isoPABA structure is expected to facilitate the hydrolysis of the hard segments by basic and acidic conditions or by the action of an esterase-enzyme. Due to the absence of adverse biological effects, the insertion of this aromatic structure would lead to high performance biodegradable materials. The synthesis route for isoPABA is depicted in Figure 2.

The synthesis of the isocyanate isoPABA has been carried out using triphosgene as a reagent and using an anhydrous toluene to avoid side-reactions, such as the formation of aromatic ureas. Indeed, the transformation of aromatic amines into isocyanates by phosgenization with triphosgene has been shown to be very effective [20]. The chemical structure of the isoPABA monomer was corroborated by NMR and ATR–FTIR spectroscopies. As shown in Figure 1, 1H-NMR spectra from diamino-PABA and isoPABA are compared.

The most relevant changes correspond to the signals located at 4.02 (NH_2_) and 6.61 ppm (aromatic). The broad signal from the amino group (NH_2_, 4.02 ppm) completely vanished after the phosgenization reaction, while the aromatic doublet (6.61 ppm) shifted to the low field (7.09 ppm) due to the electron-withdrawing nature of the isocyanate group. Additionally, the other aromatic signal slightly shifted from 7.85 to 7.97 ppm, while the signals from the propylene moiety barely changed.

Furthermore, the selective formation of the isocyanate group was confirmed by FTIR and ^13^C-NMR spectroscopies. As shown in Appendix A, the FTIR spectrum confirmed the formation of the isocyanate group due to the intense band located at 2282 cm^−1^, corresponding to the asymmetric stretching vibration mode of the isocyanate group. Moreover, the characteristic band associated with the ester group was detected at 1709 cm^−1^. The presence of this ester band confirmed the stability of this functional group under the tested conditions, since this functional group could be hydrolyzed during the phosgenization reaction by the released HCl. Similarly, the formation of the isocyanate group was also confirmed by the peaks at 125.6 and 138.1 ppm in the ^13^C-NMR spectrum, which correspond to isocyanate and aromatic carbon linked to the isocyanate group, respectively (Appendix A).

### 3.2. Synthesis and Characterization of Hard Segments Models

Segmented polyurethanes are characterized by two types of segments, hard and soft. The hard segment is built up by the reaction between a chain extender and a diisocyanate. To evaluate the characteristics of the hard segments (HS) derived from this novel aromatic isocyanate (isoPABA), in the following set of experiments, an equimolar ratio of isoPABA and short diol were combined, leading to a hard segment model polyurethane. Two types of diols were employed, namely BD and EDA-2CL. Both chain extenders exhibit high structural symmetry, which could lead to crystalline hard segments. Additionally, the EDA-2CL chain extender contains two hydrolyzable ester groups and its degradation by-products are non-toxic [12]. The synthesis of these HS model polyurethanes was carried out under standard conditions (3 h at 80 °C and rt overnight) and using anhydrous solvents to avoid secondary reactions. In both model PUs, FTIR spectra in the carbonyl stretching vibration region (1800–1650 cm^−1^) presented three major spectral components (Appendix A). The strong band at higher frequencies (1729 cm^−1^) could be assigned to “free” carbonyl groups and at lower frequencies (1710 cm^−1^) could be due to hydrogen-bonded carbonyl groups [21]. The third intensive band at 1644 cm^−1^ could be associated to amide I. Additionally, for the isoPABA–EDA-2CL HS model, the vibration of the ester group appeared overlapped at 1710 cm^−1^. Similarly, the amide II band located at 1533 cm^−1^ corresponded to the amide and urethane groups. The chemical structures of these HS model polyurethanes were confirmed by proton NMR. The signal assignment to its corresponding proton is shown in Appendix A.

The thermal transitions detected in these HS model polyurethanes were studied by DSC. In Figure 2, the second heating scan of the isoPABA–EDA-2CL derived polyurethane is shown. In this thermogram, first, a glass transition was detected at 72.4 °C from the amorphous phase of the polymer. Subsequently, two exothermic crystallization peaks and two endothermic melting peaks were detected alternately. This curve demonstrates that this HS can crystallize, with a complex behavior. The sum of the melting endotherms gives a substantially higher value than the sum of the crystallization exotherms, thus part of the HS crystallized during the fast-cooling step from the melt, offers indirect proof of the high capacity for crystallization of isoPABA–EDA-2CL HS. This crystallization could drive a better phase separation that would improve the performance of the segmented polyurethane, based on this aromatic isocyanate and this hydrolyzable chain extender.

In contrast, the BD-based HS model polyurethane (isoPABA–BD) (Appendix A) showed a glass transition at 58.2 °C and crystallization and melting peaks with similar normalized areas. Thus, this polyurethane was initially amorphous and, therefore, the crystallization capacity is lower than for isoPABA–EDA-2CL. The Tg value for isoPABA–EDA-2CL is significantly higher and could be explained by the restriction imposed by the rigid crystals present after cooling or by the stiffening effect of the amide groups or both. In addition, the crystals of isoPABA–EDA-2CL melt at a higher temperature than the crystals of isoPABA–BD.

### 3.3. Synthesis of Segmented Poly(Ester-Urethane)s

Different diols based on polycaprolactone were used as the soft segment. The selection of this polyester was motivated by the fact that it is a biocompatible and hydrolytically or enzymatically degradable material. It is well-known that the use of PCL-based polymer materials is widespread in the biomedical field, such as in scaffolds [22] or amphiphilic co-networks [23,24,25]. In addition, its main degradation product (6-hydroxyhexanoic acid) is easily transformed into adipic acid [26]. In this manner, the three building blocks of poly(ester-urethane)s can biodegrade into non-harmful side products. For the synthesis of these materials, three PCL-diols with molecular weights ranging from 500 to 2000 g mol^−1^ were selected. Typically, linear thermoplastic polyurethanes are based on macroglycols, with a molecular weight range between 1000 and 2000 g mol^−1^, which leads to good mechanical properties [27]. In this second set of experiments, the chain extenders described above were used, and also two polyurethanes based on common diisocyanates (MDI and HDI); PCL2054 and EDA-2CL, were prepared to compare with the corresponding polyurethane based on isoPABA. For all the poly(ester-urethane)s prepared, the hard segment content was set to 50% by weight, while maintaining an equimolar ratio of the NCO:OH groups. Synthesis schemes and images of these pol(ester-urethane)s are shown in Appendix A.

Regarding the solubility of the poly(ester-urethane)s, it was found that these materials were completely soluble in polar solvents, such as DMF, DMAc or DMSO, and in hexafluoroisopropanol (HFIP). The molecular weights of these synthesized polymers were determined by Size-Exclusion Chromatography (SEC). The number and weight average molecular weights and polydispersity index of these polymers are collected in Table 1. The poly(ester-urethane)s were named PUX-YZ-50 following this code: PU refers to poly(ester-urethane); X refers to the isocyanate (P = isoPABA, M = MDI and H = HDI); Y refers to the chain extender type (E = EDA-2CL and B = BD) and Z refers to the Mn (RMN) of the PCL diol (2 = 2054, 1 = 1100 and 5 = 519).

The introduction of EDA-2CL as a chain extender led to an increase in the molecular weight of their polyurethanes (entries 1–3), with respect to the classical 1,4-butanediol (entries 6–8). Indeed, the influence of the chain extender on the molecular weights was much more pronounced than the length of the soft PCL segment. In this sense, comparing PUP-2E-50 (entry 1) with PUP-2B-50 (entry 6), the molecular weight was two-fold higher, while the polydispersity remained practically unchanged; something similar was observed comparing entries 3 and 8. However, an increase in the length of the soft segment did not lead to an increase in the molecular weight of the polyurethanes.

The polydispersity values were broader for the polyurethanes bearing classical isocyanates (entries 4 and 5); however, for all the PABA-polyurethanes, the polydispersity was slightly higher than 1.5, even though they reached similar molecular weights to the MDI- or HDI-based polyurethanes (entries 4–5). GPC traces of these polymers are collected in Appendix A

### 3.4. Characterization of Segmented Poly(ester-Urethane)s

#### 3.4.1. Nuclear Magnetic Resonance (NMR)

The ^1^H-NMR spectrum of segmented polyurethane PUP-2E-50 in deuterated DMSO is shown in Figure 3.

After the insertion of the PABA moiety into the polyurethane chain, the distance between the two doublets of the AB aromatic system is considerably reduced, indeed, this distance is smaller than the isoPABA or diamino-PABA products (see Figure 1), whilst the propylene chain remains nearly unchanged. Additionally, the signal positions of the methylenes, labelled **6** (4.08 ppm) and **ε** (3.97 ppm), demonstrate that the chain extender (EDA-2CL) and the soft segment (HO-PCL-OH) have also been inserted into the polyurethane chain. Moreover, since the isocyanate group was not detected, it confirmed that the reaction was completed and the stoichiometry between the three reactants was properly adjusted.

#### 3.4.2. Infrared Analysis (FTIR–ATR)

Infrared spectroscopy is a very powerful tool for the characterization of polyurethanes, not only in the region of carbonyl groups, as discussed above, but valuable information can also be extracted through other bands. The FTIR spectra of isoPABA, PUP-2E-50 and diamino-PABA are shown in Figure 4.

In the case of isoPABA (red spectrum), the intense bands at 2280 and 1700 cm^−1^ were attributed to isocyanate (NCO) and ester (C=O) stretching vibrations, respectively. It should be noticed that the strong band of the isocyanate group completely vanished once the poly(ester-urethane) chain was formed. Furthermore, for PUP-2E-50, new absorption bands appeared at 3310 and 1638 cm^−1^, which were attributed to the stretching vibration of N–H and amide I of the urethane groups, respectively. Additionally, a single band for the N–H bond was observed in the polyurethane, compared to the three characteristic bands of the aromatic amines, such as diamino-PABA (blue spectrum), demonstrating that the insertion of the PABA motif into the poly(ester-urethane) matrix was carried out through urethane bonds. Finally, the vibrations located at 1218 and 1166 cm^−1^ were attributed to the soft phase (PCL diols), which correspond to O–C–O stretching and symmetric C–O–C stretching, respectively, of this polyester [28,29].

#### 3.4.3. Thermogravimetric Analysis (TGA)

Through this technique, the materials processing window can be known, and in the particular case of segmented polyurethanes, the chemical composition could be estimated. As urethane groups are more thermally labile than ester or amide counterparts, in many cases, the first drop is related to the thermal degradation of the hard segment, while the mass loss associated with the soft segment occurs at higher temperatures [30]. In isoPABA-based polyurethanes, the first weight lost at 276–288 °C, was only slightly higher than 50% of the weight (Figure 5B,C). All isoPABA-based polyurethanes behaved similarly, irrespective of the chain extender or the PCL length. However, polyurethanes based on the classical isocyanates, MDI (PUM-2E-50) or HDI (PUH-2E-50) (Figure 5A), showed a first weight lost much higher than the amount of the hard segment (50%).

We do not have an explanation for this difference when the initial degradation temperature is very similar for all the synthesized polyurethanes, irrespective of the diisocyanate, chain extender or PCL length (280 °C). An increase in the thermal stability of the polyurethanes when PABA motifs were incorporated has been previously reported [18], although in those polyurethanes, the PABA structure acted as a chain extender (diamino-PABA) and not as an isocyanate, and so the insertion occurred through a urea bond instead of urethane groups as in our work.

#### 3.4.4. Differential Scanning Calorimetry (DSC)

We found in the HS models that the symmetrical structure of the chain extenders (EDA-2CL and BD) and isoPABA allowed for their crystallization. Therefore, a phase separation between the different segments within the polyurethane matrix was expected. The thermal properties of these polyurethanes obtained from the DSC curves in the second heating scan are listed in Table 2.

The isoPABA-derived poly(ester-urethane)s showed a clear dependence between the PCL molecular weight and the glass transition temperature of the soft segment. In this respect, as the Mn of the polyester diol decreased, the Tg value increased from −46 to 5 °C for the EDA-2CL series (entries 1 to 3) and from −41 to 8 °C for the BD series (entries 6 to 8). This increase is related, on the one hand, to the restrictions imposed on the chain ends by the hard segments and, on the other hand, to the increase in the density of urethane groups within the polymeric matrix. Likewise, for both series, the crystallization of the soft segment was only appreciated for the highest molecular weight (PCL2054), and for the rest of the PCL molecular weights (1100 and 519 g·mol^−1^), the soft segments were amorphous.

The glass transition temperature for the polyurethanes, based on the classical isocyanates (entries 4 and 5), was slightly higher than the PCL homopolymer of 2054 g/mol [31,32], indicating that the soft segments are well separated into practically pure phases. For the polymers PUP-2E-50 and PUP-2B-50, the Tg was slightly higher, denoting that the phase separation in these polyurethanes is slightly less effective, or that the restrictions imposed by the HS crystalline phase were greater. This observation is in line with the values of the enthalpy of fusion of the crystallized soft segments. The highest enthalpies were found for the classical polyurethanes (entries 4 and 5), whilst the lowest was for PUP-2B-50 (entry 6), meaning that a higher amount of pure SS phase was present in the former and a lower amount in the latter.

All the synthesized polyurethanes showed a hard segment melting temperature that varied between 140 and 188 °C, again pointing to thermodynamic incompatibility between the segments, rendering in phase separation within the polymeric matrix. Comparing the melting temperatures of the hard segments of the polyurethanes from the aromatic isocyanates, it is observed that the PABA-derived poly(ester-urethane)s showed a higher Tm than their classical partner (PUM-2E-50). Probably, the presence of the ester groups within the isoPABA structure could increase the polar interactions within the hard segment, leading to an increase in the melting point of the hard phase. In addition, the flexibility of the propylene spacer should also be addressed, which could allow the aromatic rings to be effectively accommodated within the hard segment. However, for MDI, the methylene group between both aromatic rings could hardly provide such flexibility. This is supported by the enthalpy of fusion values (∆H_2_), which are higher for the isoPABA derivatives compared to PUM-2E-50. In the case of the polyurethane derived from the aliphatic isocyanate HDI, the enthalpy of fusion of the crystalline phase (∆H_2_) was the highest of the series, probably due to the symmetry and flexibility of the isocyanate and the absence of side groups that hinder the crystallization of this segment. Therefore, due to all these structural factors, the hard segment (HDI+EDA-2CL) of this polyurethane was the most crystalline.

#### 3.4.5. Mechanical Properties

The mechanical properties of synthetized poly(ester-urethane)s were evaluated by tensile stress–strain tests (Appendix A). The results are summarized in Table 3. Based on these results, it could be observed that there are two types of mechanical behavior due to polymer crystallinity.

On the one hand, the polyurethanes based on the shorter soft segment (PCL519) (entries 3 and 8) showed typical elastomeric behavior, with low modulus and, after rupture, the specimens practically recovered their initial shape.

On the other hand, when using longer soft segments (PCL1100 and PCL2054) (entries 1, 2, 4, 5 and 6), the polyurethanes behaved like plastics and did not recover their original shape. At low deformations, the stress grew rapidly until a yield point was reached. After this point, stress growth was lower until the break. This behavior is consistent with a co-continuous morphology, where before reaching the yield point the response was governed by the hard segment phase, while after this point the response of the material was determined by the soft segment phase until the break.

The mechanical properties depended mainly on the molecular weight of the polyurethane. As it can be seen in Table 1, when BD was used as a chain extender with isoPABA, the molecular weights were discrete, leading to poor mechanical properties. This was the case also for the polyurethane from PCL1100, isoPABA and EDA-2CL. For the other polymers, with sufficiently high molecular weights, tensile strains were high.

#### 3.4.6. Hydrolytic and Enzymatic In Vitro Degradation

The polymer samples were immersed in a phosphate buffer solution (pH = 7.4) and incubated at 37 °C and their weight changes in the hydrated state were monitored at certain time intervals. Figure 6 shows the curves of the polymers obtained from isoPABA and EDA-2CL as hard segments (PUP-5E-50, PUP-1E-50 and PUP-2E-50). For all the studied polymers, weight change curves presented two differentiated states. At the beginning of the immersion, a maximum water absorption was reached, followed by a very slow decay of weight over time. This behavior was independent of the polymer studied. The shape of the weight loss curves in the swollen state is characteristic of surface erosion, with very low total degradation.

The maximum water absorption showed a relationship with the molecular weight of the soft segment, with the lower molecular weights (PUP-5E-50 and PUP-1E-50) reaching higher maximum absorptions than PUP-2E-50. This behavior would be due to the fact that the crystalline fraction, presented in PUP-2E-50, would hinder the inclusion of water within the polymeric structure, whereas the completely amorphous character of the SS in the polyurethanes based on PUP-5E-50 and PUP-1E-50 favors water access. This trend is in line with the DSC results described above and has been previously reported [33,34]. For all the cases studied, the maximum absorption occurred practically within the first 24 h of immersion and subsequently, the weight of the polymer decreased progressively.

Regarding the nature of the isocyanate (Appendix A), it was observed that the maximum water absorption was slightly higher for the isoPABA-polyurethane (PUP-2E-50, 2.81%) compared to the classic isocyanates, HDI (PUH-2E-50, 2.18%) and MDI (PUM-2E-50, 2.16%). This increase in water absorption could be promoted by the polarity conferred by the ester groups present within the structure of the novel aromatic isocyanate. Similarly, the nature of the chain extender had a similar effect (Appendix A) because the polar (amide) groups provided by EDA-2CL would increase hydrophilicity through hydrogen bonding with water molecules, leading to increased water adsorption. Comparing both effects, it was appreciated that the nature of the chain extender was more relevant with respect to the effect provided by the isocyanate, since the maximum absorption of PUP-2E-50 amounted to 2.81%, while for PUP-2B-50, it was practically not exceeding 1.50%.

Globally, maximum water absorption was very low in these polyurethanes, ranging from 1.6 to 4.3%.

In the second stage of film degradation, the degradation rate was slightly higher for polymers with a higher swelling index, i.e., polymers based on isoPABA, EDA-2CL or with short PCL diol. This could be caused by the fact that the access of water to the hydrolyzable groups is less hindered, and the degraded products diffused more easily.

Subsequently, the enzymatic degradation of these polymers was also evaluated gravimetrically using the porcine liver esterase (PLE) enzyme.

As shown in Table 4, after 7 days of incubation in absence of an enzyme, the PABA-based polyurethanes showed a small swelling between 1.5 and 3%. However, in the presence of the PLE, the degradation of the polyurethanes is more marked. The enzyme PLE promotes the hydrolysis of ester bonds contained in the soft segment. This effect is smaller on the polyurethane with a longer PCL (PUP-2E-50) because the crystalline fraction is more resistant to enzymatic degradation, while the fully amorphous polymers (PUP-1E-50 and PUP-5E-50) present a molar ratio of PCL higher than the longer counterpart and, therefore, a higher number of hydrolysis points. Different levels of degradation are reported depending on the molecular weight, crystalline or amorphous content, or the presence of C–C strong bonds, which are resistant to attack by enzymes [35,36].

#### 3.4.7. Cytotoxicity Test

Toxicity assays were developed in the absence and presence of PLE to study the toxicity of the degradation products. Figure 7 shows the cytotoxicity on HOb and HC-a cells.

PABA-based polyurethanes showed a small but significant toxicity on both HOb and HC-a cells, depending on the molecular weight of the PCL. Low molecular weight PCL based-polyurethanes (PUP-5E-50) did not show a significant toxicity on HOb but started to show significant differences on HC-a cells. For medium and high molecular weight PCL (PUP-1E-50 and PUP-2E-50), cell viability reached close to 80%, being independent of the presence or absence of PLE. Despite the fact that aromatic diisocyanates are mentioned as cytotoxic, in this work, the contents of aromatic diisocyanates presented no cytotoxic results; other works also reported non mutagenic behavior for polyurethanes obtained with this type of diisocyanates [37].

#### 3.4.8. Adhesion and Proliferation Assays

Adhesion and proliferation assays were made using Alamar Blue tests. Figure 8 shows the adhesion and proliferation of HOb and HC-a cells on poly(ester-urethane) films at 1, 7 and 14 days.

All the PUs allow the adhesion and proliferation of HOb and HC-a cells. The lower adhesion in comparison with the control is due to the different topography and hydrophilicity of the PUs; cells need time to produce extracellular matrix to adhere on the surface. The initial adhesion of the cells is significantly higher for polyurethanes with high molecular weight PCL (PUP-2E-50). This could be due to the differences in the elastic modulus of the systems; for example, HOb cells present higher adhesion efficiency for higher elastic modulus [38]. Cell proliferation increases over the time but depends on the PCL molecular weight. Proliferation for polyurethanes with high molecular weight PCL (PUP-1E-50 and PUP-2E-50) is lower than PUP-5E-50. This could be correlated with the results in Figure 7, where PUP-2E-50 showed a significant toxicity. This increase in the toxicity also has an effect on the proliferation of the cells on the polyurethanes.

Other studies on polyurethanes reported that ≥70% of cell viability is promising for biological applications [39]. These good results in cell adhesion and toxicity make this kind of novel polyurethane a good option to building extracellular matrices. In addition, the mechanical properties of these systems are tunable by the election of different soft and hard segments. There are many reports of polyurethanes with tissue engineering applications [40,41]. The literature mentioned that the rupture of ester linkages increases hydrolytic and enzymatic degradation. In addition, using molecules such as esterases increments the rate of degradation in polyurethanes. Aliphatic hard segments have been chosen in order to avoid cytotoxic degradation products, but there are studies with aromatic molecules, such as polyaniline, used to obtain polyurethanes with self-healing behaviors, that result in non-cytotoxic polyurethanes, which are shown to have excellent cellular adherence [42]. In this work, an aromatic diisocyanate (isoPABA) was prepared from aromatic diamine, that is reportedly not cytotoxic or mutagenic, according to the MTT results. On the other hand, the Alamar Blue results show good cellular adhesion and proliferation for the PUs.

## 4. Conclusions

A series of polyurethanes bearing p-aminobenzoate units were prepared through the synthesis of the novel symmetric aromatic isocyanate isoPABA. This diisocyanate led to polyurethanes with high molecular weights when the hydrolyzable chain extender EDA-2CL was used. The chemical structures of the biocompatible segmented polyurethanes were confirmed by several techniques (FTIR–ATR and NMR).

As shown by TGA, although the urethane groups are the thermally weakest linkages, the thermal stability of all the polymers presented was sufficiently high, with a wide processing window.

Due to the symmetrical structure of the novel isocyanate and the flexibility introduced by its propylene spacer, the hard segments of the polyurethanes crystallized easily, favoring phase separation within the poly(ester-urethane) matrix. However, the enthalpies of fusion of the crystallized soft segments were lower for the PABA-polyurethanes than those of their classical counterparts, which was probably due to a lower fraction of pure HS. Although the melting enthalpies of the isoPABA-based HS increased with respect to its aromatic MDI partner, the aliphatic isocyanate HDI was shown to be the most crystalline HS.

Soft segments derived from low molecular weight PCL (PCL519) were completely amorphous, and the corresponding polyurethanes elastomeric, recovering the initial shape after rupture. Poly(ester-urethane)s based on PCL1100 and PCL2054 showed the behavior of plastics. The polyurethanes bearing the hydrolyzable chain extender reached higher molecular weights than the BD-polyurethanes, achieving better performance and mechanical properties.

For all the poly(ester-urethane)s bearing isoPABA motifs and EDA-2CL chain extenders, cell viability after 28 days was higher than 80%, demonstrating that these polymers were non-toxic. Moreover, Alamar Blue results indicated a good cell adhesion and cytotoxicity on the surfaces of these non-toxic biodegradable polyurethanes. Ultimately, the diisocyanate isoPABA is an attractive moiety for the preparation of poly(ester-urethane)s to build extracellular matrices.

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
