# Peer review of "Biodegradable and Biocompatible Thermoplastic Poly(Ester-Urethane)s Based on Poly(ε-Caprolactone) and Novel 1,3-Propanediol Bis(4-Isocyanatobenzoate) Diisocyanate: Synthesis and Characterization"

_polymers, 2022, doi:10.3390/polym14071288_

Round 1

Reviewer 1 Report

  1. Abstract is too general. It should provide more information about the tests that have been carried out.
  2. No keywords.
  3. The Introduction section is too general. The authors provide generally known information and should focus on similar research and present it in this section.
  4. The method of citing literature is inconsistent with the requirements of the journal. References should be in square brackets and not superscript.
  5. Materials and Methods section. The authors should characterize the suppliers of raw materials for the research in more detail.
  6. NMR spectra (Figure 1) should be made the same as in Supplementary files.
  7. Conclusion section should be more supported by research results, not just general statements.
  8. The References section is written in a different font and has a different font size than the requirements for the reference manuscript.

Author Response

Reviewer 1

  • Abstract is too general. It should provide more information about the tests that have been carried out.

We appreciate the valuable comment. Following his/her comment, we summarised the results reported in the manuscript. Introducing the next lines:

“The behaviour of short PCL-based polyurethane was elastomeric, whilst increasing the molecular weight of the soft segment led to plastic polyurethanes. Water uptake was hindered for long PCL due to the crystallisation of the soft segment within polyurethane matrix. Furthermore, two different types of chain extender, hydrolysable and non-hydrolysable, were also evaluated: polyurethanes based on hydrolysable chain extender reached higher molecular weights, thus leading to better performance than their unhydrolysable counterparts.”

  • No keywords.

We thank the reviewer for feedback. The following keywords have been included in the revised version of this manuscript: Biodegradability; biocompatibility; thermoplastic polyurethane; non-toxic polyurethane; hydrolysable chain extender.

  • The Introduction section is too general. The authors provide generally known information and should focus on similar research and present it in this section.

We thank the reviewer for his/her input. However, we believe that the structure of the introduction is concrete and straightforward. The first paragraph gives a brief introduction on how polyurethanes are prepared to understand the two-phase structure (hard and soft segment). The presence of polyurethanes is then addressed only in the biomedical sector, discarding other sectors that would be outside the framework of this manuscript. Subsequently, the problem of polyurethanes in this sector is described, i.e. the toxicity of aromatic amines due to the use of aromatic isocyanates. For this reason, the following paragraph presents the alternatives proposed—the use of aliphatic or amino acid isocyanates. The main drawbacks (aliphatic character and asymmetry) of these alternatives are then described, leading to polyurethanes with poorer performance than their aromatic counterparts. In the final part of the introduction an aromatic and symmetrical isocyanate that allows the synthesis of polyurethanes without toxicity is presented, on account of the compatibility of the amine used.

  • The method of citing literature is inconsistent with the requirements of the journal. References should be in square brackets and not superscript.

In the revised version of the manuscript, the style of the references has been adjusted to the style defined by the journal.

  • Materials and Methods section. The authors should characterize the suppliers of raw materials for the research in more detail.

The authors appreciate the observation made by the reviewer. The raw material suppliers have been updated, in order: company brand, city and country.

  • NMR spectra (Figure 1) should be made the same as in Supplementary files.

Acknowledging the comment and following the reviewer's recommendation, the assignment of the NMR signals for both products has been included in the new figure 1. 

  • Conclusion section should be more supported by research results, not just general statements.

We are grateful for the reviewer's feedback. Following his/her recommendations, we have included the following conclusions:

    • As shown by TGA, although the urethane groups are the thermally weakest linkages, the thermal stability of all polymers presented was sufficiently high, with a wide processing window. Due to the symmetrical structure of the novel isocyanate and the flexibility introduced by its propylene spacer, the hard segments of the polyurethanes crystallised easily, favouring phase separation within the poly(ester-urethane) matrix. However, the enthalpies of fusion of the crystallized soft segment were lower for PABA-polyurethanes respect to their classical counterparts, which was probably due to a lower fraction of pure HS. Although the melting enthalpies of the isoPABA-based HS increased with respect to its aromatic MDI-partner, the aliphatic isocyanate HDI was shown to be the most crystalline HS.

    • The polyurethanes bearing the hydrolysable chain extender reached higher molecular weights than the BD-polyurethanes, achieving better performance and mechanical properties.

  • The References section is written in a different font and has a different font size than the requirements for the reference manuscript.

As discussed above, the format of the references has been updated following the style defined by the journal.

Reviewer 2 Report

Comments to the authors

Professor Ángel Marcos-Fernández's group has presented a very interesting piece of research work based on the Biodegradable and Biocompatible Thermoplastic Poly(ester- 2 urethane)s Based on Poly( -Caprolactone) and Novel 1,3- 3 Propanediol Bis(4-Aminobenzoate) Diisocyanate. Synthesis and Characterization. in this work researchers have synthesized polyurethane and two different types of chain extender, hydrolysable and non-hydrolysable, were evaluated, showing significant differences in material performance, and showed good cell adhesion and cytotoxicity with human osteoblast.

I strongly recommend revising this manuscript as follows:

  1. In the introduction, the part author should discuss other relative crosslinked biodegradable materials such as amphiphilic materials APCN based on polycaprolactone.  https://doi.org/10.1016/j.polymer.2016.07.033, https://doi.org/10.1021/acsabm.8b00461 https://doi.org/10.1039/9781788015769-00047  author should include information about the above-suggested information in this manuscript.
  2. The author should include information about the solubility of the newly synthesized polymer.
  3. The Picture of prepared films should be included in this manuscript.
  4. The AFM analysis of the prepared films should be included in this manuscript.
  5. The author should add the swelling result of the film in graphical format, and a time-wise swelling experiment test should be performed in different pH such as 8, 7.4, 5 and the result should add in this manuscript.
  6. The confocal images of cell viability and adhesion experiment should be added in this manuscript.
  7. The GPC traces of different polymers should be added to the manuscript
  8. In line no 212 correct as 1.4 x 105
  9. The all-synthetic reactions scheme should be gathered at one place in the manuscript with prepared polymeric films image in one figure that will be helpful for the readers.
  10. The author should organize the mechanical strength data in the graphical format instead of the table.

Author Response

Professor Ángel Marcos-Fernández's group has presented a very interesting piece of research work based on the Biodegradable and Biocompatible Thermoplastic Poly(ester- 2 urethane)s Based on Poly( -Caprolactone) and Novel 1,3- 3 Propanediol Bis(4-Aminobenzoate) Diisocyanate. Synthesis and Characterization. in this work researchers have synthesized polyurethane and two different types of chain extender, hydrolysable and non-hydrolysable, were evaluated, showing significant differences in material performance, and showed good cell adhesion and cytotoxicity with human osteoblast.

I strongly recommend revising this manuscript as follows:

  • In the introduction, the part author should discuss other relative crosslinked biodegradable materials such as amphiphilic materials APCN based on polycaprolactone.
    https://doi.org/10.1016/j.polymer.2016.07.033
    https://doi.org/10.1021/acsabm.8b00461
    https://doi.org/10.1039/9781788015769-00047
    author should include information about the above-suggested information in this manuscript.

Following the indications provided by the reviewer, the references mentioned have been included in the following sentence:

It is well-known, PCL-based polymer materials are widespread in the biomedical field, such as scaffolds [22] or amphiphilic co-networks.[23-25]

  • The author should include information about the solubility of the newly synthesized polymer.

We thank the reviewer for his/her input to improve the scientific quality of the manuscript. Although the solubility of the polymers was already indicated in the initial version, we have considered it appropriate to introduce the following sentence:

Regarding the solubility of the poly(ester-urethane)s, it was found that these materials were completely soluble in polar solvents, such as DMF, DMAc or DMSO and in hexafluoroisopropanol (HFIP)

  • The Picture of prepared films should be included in this manuscript.

We appreciate the reviewer's comment, however, we believe that the pictures of the polyurethane films should be present in the supplementary material, instead of the main text. Additionally, as reviewer proposed later, we have also included the reaction scheme for the preparation of these polyurethanes in the same figure. (figure S7).

  • The AFM analysis of the prepared films should be included in this manuscript.

The authors are grateful for the reviewer's comment. Regarding this issue, the morphological analysis of our materials, through AFM or SEM, is not possible because their surfaces are completely flat and no valuable information can be extracted. As an example, we attach three SEM images of the surfaces of three different polymers. As can be seen, the surfaces are quite similar and the composition is different. Something similar would be the case with AFM analysis.

The image is in the attachment file.

  • The author should add the swelling result of the film in graphical format, and a time-wise swelling experiment test should be performed in different pH such as 8, 7.4, 5 and the result should add in this manuscript.

We are grateful for the reviewer's valuable note. The polymer swelling plots were initially included in the manuscript (figure 6) and in the supplementary material (figures S10 and S11). Regarding the study at different pH, we would like to comment that your proposal is indeed very interesting but would currently be outside the framework of the present manuscript. Basically, the selection of pH=7.4 is based on the fact that it is the most commonly used medium for preliminary biological studies. The possibility of studying the behaviour of our materials at different pH remains of high interest, and will be taken into account for our future research.

  • The confocal images of cell viability and adhesion experiment should be added in this manuscript.

Regarding the confocal images, we would like to indicate that Cell viability, adhesion and proliferation assays was carried out with quantitative test: MTT for cell viability and Alamar Blue for cell adhesion and proliferation. These assays are based on the methabolic process of the cells and they did not provide any fluorescence in the cell. for this reason, therefore is not possible to visualized the cells by confocal microscopy using this tests.

  • The GPC traces of different polymers should be added to the manuscript

We appreciate the reviewer's recommendation and traces of the GPC results have also been included within the supplementary material. (figure S8)

  • In line no 212 correct as 1.4 x 105

Following the reviewer's recommendation, the formats of these numbers have been adjusted.

  • The all-synthetic reactions scheme should be gathered at one place in the manuscript with prepared polymeric films image in one figure that will be helpful for the readers.

We appreciate the reviewer’s comment. Following his/her recommendation and his/her third comment, we included the reaction scheme in the same figure. (Figure S7)

  • The author should organize the mechanical strength data in the graphical format instead of the table.

We would like to comment to the reviewer of the manuscript that the graphs of the mechanical properties were initially included in the supplementary material. In the main text of the manuscript, we decided to include the summary table to facilitate the interpretation of the results for potential readers of this work.

Round 2

Reviewer 1 Report

The article has been significantly improved. I think it may be published in Polymers.